# Assessment of Polydopamine to Reduce *Streptococcus mutans* Adhesion to a Dental Polymer

**DOI:** 10.3390/pathogens12101223

**Published:** 2023-10-08

**Authors:** Santiago Arango-Santander, Carlos Martinez, Claudia Bedoya-Correa, Juliana Sanchez-Garzon, John Franco

**Affiliations:** 1GIOM Group, Faculty of Dentistry, Universidad Cooperativa de Colombia, Medellin 055421, Colombia; carlos.martinez@campusucc.edu.co (C.M.); claudia.bedoyac@campusucc.edu.co (C.B.-C.); john.francoa@campusucc.edu.co (J.F.); 2Faculty of Dentistry, CES University, Medellin 050021, Colombia; jsanchezg@ces.edu.co; 3Salud y Sostenibilidad Group, School of Microbiology, Universidad de Antioquia, Medellin 050010, Colombia

**Keywords:** *Streptococcus mutans*, polydopamine, surface modification, surface coating, biomimetics, antibacterial effect, poly(methyl methacrylate)

## Abstract

Bacterial adhesion to the surface of materials is the first step in biofilm formation, which will lead to conditions that may compromise the health status of patients. Recently, polydopamine (PDA) has been proposed as an antibacterial material. Therefore, the objective of the current work was to assess and compare the adhesion of *Streptococcus mutans* to the surface of poly(methyl methacrylate) (PMMA) discs that were modified using PDA following a biomimetic approach versus smooth PDA-coated PMMA surfaces. In addition, an assessment of the growth inhibition by PDA was performed. PMMA discs were manufactured and polished; soft lithography, using the topography from the *Crocosmia aurea* leaf, was used to modify their surface. PDA was used to smooth-coat PMMA discs by dip-coating. The growth inhibition was measured using an inhibition halo. The surfaces were characterized by means of atomic force microscopy (AFM), the contact angle (CA), and Fourier-transform infrared spectroscopy (FTIR). Polydopamine exhibited a significant antibacterial effect when used directly on the *S. mutans* planktonic cells, but such an effect was not as strong when modifying the PMMA surfaces. These results open the possibility of using polydopamine to reduce the adhesion and growth of *S. mutans*, which might have important consequences in the dental field.

## 1. Introduction

Poly(methyl methacrylate) (PMMA) is a polymer that is industrially used for the fabrication of a great variety of biomedical appliances used every day. In the field of dentistry, PMMA is used to fabricate different prosthetic and orthodontic appliances. Among its advantages, the low inflammatory reactions, its biocompatibility, its high resistance to UV light, its high resistance to chemical attacks, its easiness to use, and the low cost are the most representative [1]. However, its high susceptibility to bacterial adhesion is one of its most relevant disadvantages [2].

*Streptococcus mutans* is one of the main pathogens playing a significant role in the etiology of oral diseases, including caries and peri-implantitis, since it produces a biofilm that adheres to natural and artificial surfaces, such as composites, amalgam, and other dental biomaterials [3,4]. Such adhesion to the surface of biomaterials is influenced by the composition of the biomaterial and the surface properties, including the roughness and hydrophobicity [5]. A high roughness [6,7], associated with the long exposure periods of biomaterials in the oral cavity, will lead to an increase in the adhesion of normal inhabitants from the oral microbiome, thus forming a biofilm [8]. This resulting biofilm might produce favorable conditions for the onset of oral pathologies, including caries and peri-implantitis, among many others [9].

Numerous strategies have been proposed to reduce the bacterial adhesion and growth on the surface of biomaterials [10,11], including dental hygiene therapies, diet modifications, and the use of topical fluoride [12]. However, these methods are highly dependent on the patient, which might reduce their efficiency. Current investigations have focused on assessing other strategies that involve biomaterials, and surface modification, both chemical and physical, has been one of the most addressed in the last decade [13,14,15]. In this regard, many chemical approaches have been proposed, including the use of polydopamine (PDA) as an antibacterial agent [16]. PDA is a biological neurotransmitter that is present in numerous living organisms and may be found in high quantities in marine mussels. This adhesive protein is characterized by a high presence of polyphenols with a catechol group, which is released in the shape of a filament and polymerizes by oxidation [1,17,18,19], although many details of the structure and formation are still being actively investigated.

Surface modification may be carried out by following a wide variety of techniques [20,21], including soft lithography. These series of techniques are based on copying and transferring the features from a master model to another surface by means of a poly(dimethyl siloxane) (PDMS) stamp. This polymeric material has the ability to copy three-dimensional, curved structures; it is compatible with a wide array of materials, and it is easy to use [22]. The master model may be obtained through different methods and from many sources, with nature being the inspiration to use the topography from natural surfaces on artificial materials [23,24], an approach known as biomimicry. Many natural surfaces have been used to modify artificial materials, including animal surfaces, such as sharkskin and cicada wings [25], and vegetal sources, such as taro and montbretia leaves [26]. These surfaces exhibit particular characteristics that are useful for modifying artificial materials, such as a high hydrophobicity and self-cleaning properties [27].

Despite the fact that polydopamine has shown antibacterial properties, the use of this chemical compound alone as part of a physico-chemical surface modification strategy, without the use of other chemical compounds, to reduce the adhesion and growth of a bacterial species has been scarcely reported in the literature [28]. Therefore, the objective of this work was to assess and compare the adhesion of *S. mutans* to the surface of PMMA discs that were modified using PDA following a biomimetic approach versus smooth PDA-coated PMMA surfaces. In addition, an assessment of the growth inhibition of this bacterial species was performed by placing it in close contact with PDA.

## 2. Materials and Methods

### 2.1. Substrates

Self-polymerization PMMA (D3 shade, Veracril, New Stetic, Guarne, Colombia) discs of 10 mm in diameter and 1 mm in thickness were fabricated. Sequential polishing using silicon carbide papers (800 to 1500 grit, Abracol, Girardota, Colombia) was performed, followed by cleaning with 70% ethanol (Merck KGaA, Darmstadt, Germany). Sterilization was carried out by subjecting the discs to UV light (Accu-clean, Effitech S.A.S., Medellín, Colombia) for 15 min.

### 2.2. Master Model and Soft Lithography

The master model used in this investigation was the leaf of montbretia (*Crocosmia aurea*). This leaf was selected based on its high hydrophobicity and self-cleaning ability. Leaf fragments of 3 cm × 3 cm were cut and attached to the bottom of a silicon container. To fabricate the stamps, PDMS (Silastic T-2, Dow Corning Corporation, Midland, MI, USA) was prepared according to the instructions and poured into the containers to cover the leaf fragments. The silicon containers were placed under a vacuum to remove air bubbles and left to polymerize for 24 h. A subsequent thermal treatment at 80 °C for 3 h was performed to finish the polymerization process. 

### 2.3. PDA Preparation

PDA synthesis was carried out by following the manufacturer’s instructions. Briefly, dopamine (3-hydroxitiramine chlorhydrate, dopamine chlorhydrate cat. #H8502, Sigma-Aldrich, St. Louis, MO, USA) was dissolved in 10 nM of Tris-HCl buffer (Tris—Bio Basic, Markham, ON, Canada; HCl—Merck KGaA) at a pH of 8.5. Two concentrations (2 mg/mL and 10 mg/mL) were obtained. 

### 2.4. Surface Preparation

The PMMA discs were divided into five experimental groups. The PMMA discs in group I were only polished (control group); the discs in group II were dip-coated with the 2 mg/mL PDA solution. The discs were immersed in the PDA solution for 60 min under continuous agitation at RT. Then, the discs were removed from the solution and let dry for 20 min in a vertical laminar airflow chamber (BioBase, Jinan, China). For the discs in group III, the PDMS stamps were immersed in the 2 mg/mL PDA solution for 60 min under continuous agitation at RT. After this period, the stamps were placed in the vertical laminar airflow chamber for 20 min. Then, the stamps were carefully placed on the surface of the PMMA discs and gentle pressure was applied to transfer the topography from *C. aurea* to the surface of the PMMS discs. The discs in group IV were dip-coated with the 10 mg/mL PDA solution as described for group II and the discs in group V were treated the same as those in group III, but with the 10 mg/mL PDA solution.

### 2.5. S. mutans Inoculum Preparation

For microbiological testing, *S. mutans* (ATCC 25175, American Type Culture Collection, Manassas, VA, USA) was used. *S. mutans* was grown in BHI agar (Difco Laboratories, Saint-Ferréol, France) and incubated at 37 °C for 18 h under microaerophilic conditions (5% CO_2_ atmosphere). After incubation, bacterial suspensions were prepared by seeding bacterial cultures into 10 mL of BHI broth (Merck KGaA). Then, the turbidity conditions were measured using a turbidity meter (Velp Scientifica, Usmate Velate, Italy) until a turbidity of 90 ± 5 NTUs (nephelometric turbidity units), corresponding to a bacterial concentration of 1.5–2.0 × 10^8^ CFU/mL (colony-forming units), was obtained. 

### 2.6. Antibacterial Effect of PDA on S. mutans

In order to test the possible synergistic effect of PDA on the surface modification of PMMA discs, its antibacterial effect on *S. mutans* was evaluated. The protocols by Jiang et al. [29] and Airen et al. [30] were followed with minor modifications. Briefly, a sample of the bacterial suspension was taken with a sterile cotton swab and seeded on Mueller–Hinton agar (Merck KGaA). Wells of 7 mm in diameter on each agar plate were cut and 50 µL of PDA at different concentrations (2, 4, 6, 8, and 10 mg/mL) was independently added. A positive control was performed with 50 µL of 0.2% chlorhexidine (Clorhexol, Farpag S.A.S., Bogotá, Colombia) and a negative control was performed with 10 nM Tris-HCl buffer at a pH of 8.5. The cultures were allowed to repose for 15 min at RT to facilitate PDA diffusion and were then incubated under microaerophilic conditions (5% CO_2_) at 37 °C for 24 h. After incubation, the diameter of the inhibition zone was measured. The assays were performed in triplicate.

### 2.7. Surface Characterization

The polished and modified PMMA surfaces were characterized by their hydrophobicity and roughness. The hydrophobicity was evaluated using the contact angle (CA) method by following the sessile drop technique. A drop of 4 µL of saline solution (Corpaul, Guarne, Colombia) was added to the surfaces of five discs from each group. A camera with a 25× macro lens (CoPedvic, Shenzhen, China) was used to obtain images, and the contact angle was determined using the AxioVision software (v4.9.1.0). For the roughness evaluation, images of 50 µm × 50 µm were obtained with an atomic force microscope (AFM, XE7, Park Systems, Suwon, Republic of Korea) in contact mode and the arithmetic average of the roughness profile (Ra) was calculated using the XEI software V. 4.1.0 (XE7, Park Systems). AFM images of 5 µm × 5 µm for the *C. aurea* leaf, the control group, and experimental groups II and V were obtained to observe the topography of each surface. The AFM images of the *C. aurea* leaf and the control surface were obtained in tapping mode, while the images of experimental groups II and V were obtained in contact mode. The images were analyzed using the Gwyddion software (v2.53, Czech Metrology Institute, Jihlava, Czech Republic).

### 2.8. Spectroscopic Analysis

Fourier-transform infrared spectroscopy (FTIR) spectra were obtained for dopamine hydrochloride, PMMA, and the 10 mg/mL PDA-coated PMMA. The spectra were recorded with a spectrometer (Shimadzu IRTracer-100, Kyoto, Japan) in the wavenumber range of 400–4000 cm^−1^.

### 2.9. S. mutans Adhesion Testing

PMMA discs from the experimental and control groups were placed inside 24-well polystyrene plates (Costar, Corning Inc., Corning, NY, USA). A total of 100 µL of the bacterial suspension was added to the surface of each disc. The polystyrene plates were incubated under microaerophilic conditions (5% CO_2_) for 6 h to allow bacterial adhesion. Then, the plates were removed from the incubator and each disc was washed with a 0.9% sterile saline solution three times to remove unattached or poorly adhered bacteria. Then, each disc was sonicated in 2 mL of a 0.9% saline solution using an ultrasonic sonicator (QSonica Q500, Newtown, CT, USA) for 3 s at 50% power to detach the adhered bacteria. Dilutions of 10^−1^–10^−3^ were carried out from the sonicated products and 10 µL from each dilution was seeded on BHI agar by following the drop method. The cultures were incubated under microaerophilic conditions (5% CO_2_) for 48 h and the CFUs were counted. The assays were performed in triplicate.

### 2.10. Staining of S. mutans Adhered to PMMA Discs

Viable *S. mutans* cells adhered to the PMMA disc surfaces were characterized using the LIVE/DEAD *Bac*Light™ Bacterial Viability Kit (Thermo Fisher Scientific, Waltham, MA, USA). Syto 9 and propidium iodide (PI) were prepared by following the manufacturer’s instructions. In summary, 3 µL of each staining agent was mixed in 1 mL of 0.9% saline (Corpaul) in a microcentrifuge tube that was kept in ice during the procedure. After the incubation period with the *S. mutans* suspension, the experimental and control PMMA discs were washed with 100 µL of 0.9% saline (Corpaul) to remove unattached or poorly adhered bacteria. A total of 30 µL of the staining solution was added to the surface of each disc and incubation for 15 min in the dark was allowed. The PMMA discs were then washed with 30 µL of 0.9% saline (Corpaul) to remove the excess staining solution and were placed on a glass coverslip (24 × 60 mm and 0.13–0.17 mm thick). The PMMA surfaces were observed using an inverted microscope (Motic AE31E, Xiamen, China) equipped with a fluorescence attachment, an FTIC filter (EX at 480/30× EM at 535/40 m), and a Texas Red filter (EM at 560/40× EM at 635/60 m) at 100× Viable cells with intact membranes were stained green and cells with compromised membranes were stained red. Images were captured using a camera (Moticam ProS5 Lite, Motic) and software (Motic Image Plus V3.1). Image processing was performed using the ImageJ software V.1.53t [31].

### 2.11. Statistical Analysis

A descriptive analysis through the estimation of summary measures (central tendency and dispersion or position measures) for the roughness (µm), CFU/surface count, and inhibition haloes (mm) was performed. A comparison of the results for adhesion and roughness was performed using an ANOVA test and a multiple comparison analysis using Tukey’s HSD test when the null hypothesis was rejected. A comparison of the inhibition results with different concentrations of PDA was performed using the Kruskal–Wallis test and multiple comparisons using the Mann–Whitney U test, adjusting the statistical significance level with the Bonferroni test. The normal distribution assumption was previously verified for all the tests using the Shapiro–Wilk test. The statistical analyses were carried out using the IBM^®^ SPSS v26 software, and a *p* < 0.05 value was selected as the criterion to accept or reject the null hypothesis.

## 3. Results

### 3.1. Surface Characterization

The hydrophobicity was calculated using the CA method and the surface hydrophobicity from the *C. aurea* leaf was obtained and compared with the experimental and control surfaces. The average CA of the leaf was significantly higher than that of the experimental and control surfaces (Table 1 and Figure 1). However, the PMMA discs from groups II and IV (dip-coated in PDA at different concentrations) showed statistically significantly lower CA values than the discs from groups III and V (PDA-patterned). These last two exhibited statistically significant differences between them (*p* < 0.001), with the CA shown by group V being higher. The CA value for group I (control) was higher than the CA values for groups II and IV, but lower than those for groups III and V.

Figure 2 shows the surface roughness of the PMMA discs and AFM images of the *C. aurea* leaf, the control group, and two experimental groups. The roughness was similar in all the tested groups. No statistically significant differences were found among the control and experimental groups (*p* = 0.623).

### 3.2. Spectroscopic Analysis

The FTIR spectrum (Figure 3) obtained for dopamine hydrochloride showed peaks at the 3300 cm^−1^ region, near the 1500 cm^−1^ region, and the 1200–1400 cm^−1^ region. The PMMA showed peaks at the spectral ranges of 700–900 cm^−1^, 1000–1500 cm^−1^, 1600–1800 cm^−1^, and 2800–300 cm^−1^. The PDA-coated PMMA showed peaks consistent with PMMA and dopamine.

### 3.3. Antibacterial Effect of PDA against S. mutans

The results of the *S. mutans* exposure to increasing concentrations of PDA (2 mg/mL to 10 mg/mL) showed statistically significant differences in the diameter of the inhibition haloes between the 2 mg/mL concentration and the 8 mg/mL and 10 mg/mL concentrations (*p* = 0.020 and 0.001, respectively). Likewise, the inhibition haloes obtained with the 4 mg/mL concentration were significantly smaller than those obtained with the 10 mg/mL concentration (*p* = 0.020). No statistically significant differences were found when the inhibitory effect of chlorhexidine and all the tested PDA concentrations were compared (Figure 4 and Figure 5).

### 3.4. Adhesion of S. mutans to the Surface of PMMA Discs

After the incubation period, the amount of viable *S. mutans* cells adhered to the surface of PMMA discs was determined by counting the CFUs per surface (CFU/surface). Significant differences were found between group I and group IV (*p* = 0.004), with the CFU/surface being higher for group IV. Similar results were found when comparing the results for groups I and II (*p* = 0.048, Figure 6).

DEAD/LIVE staining confirmed the results obtained after counting the CFUs per surface (Figure 7).

The polished PMMA showed predominantly live bacteria (Figure 7a), while more dead bacteria were observed in the PMMA discs that were smooth-coated and patterned with 2 mg/mL PDA (Figure 7b,c) or 10 mg/mL PDA (smooth-coating (Figure 7d) and patterning (Figure 7e)).

## 4. Discussion

Recently, investigations into the surface modification of biomaterials have rendered a variety of alternatives to improve some surface characteristics with the aim of reducing bacterial adhesion. Among such alternatives, the use of PDA has been investigated due to its high adhesive properties and biocompatibility, among other factors [16]. Different authors have addressed the use of PDA to modify the surface of materials to reduce bacterial adhesion [32]. However, most investigations have not made use of PDA alone, but as a compound to anchor other chemical antibacterial agents. Liu et al. [28] coated the surface of zirconia discs with PDA and evaluated the adhesion of *Streptococcus gordonii* and *S. mutans*, and found a significant reduction in the adhesion of both. Choi et al. [33] loaded PDA with silver and used it to coat titanium specimens. Then, they tested the antibacterial effect on *S. mutans* and *Porphyromonas gingivalis* and found a reduction in both the colony formation and the growth for both strains. Hu et al. [34] grafted a Ti_6_Al_4_V alloy with PDA and then coated it with a graphene oxide/zinc oxide (GO/ZnO) nanocomposite coating to test the antibacterial activity against *Escherichia coli* and *Staphylococcus aureus*, and found a strong effect against these bacterial species. 

Physical surface modification has been demonstrated as an approach to reduce bacterial adhesion to surfaces [35,36,37]. Xu and Siedlecki [38] modified the surface of polyurethane to expose it to *Staphylococcus epidermidis* and *S. aureus*, which are bacterial species associated with infections produced by implanted medical devices. A significant reduction in the adhesion of both species was found. In addition, they found that biofilm formation was inhibited from 2 to 5 incubation days on modified surfaces, which made this technique suitable for preventing streptococcal infections associated with biomaterials. Chung et al. [39] found a reduction in the adhesion of *S. aureus* to the surface of a biomimetically modified polymer using a model of sharkskin. 

The current work attempted to combine both strategies, namely biomimetic physical surface modification and the chemical use of PDA, as a synergistic approach to modify the surface of dental PMMA. The results from the FTIR evaluation for dopamine hydrochloride showed peaks at 3346 cm^−1^ (O–H asymmetric stretching vibration) [40], 1618 cm^−1^ (amide I), 1502 cm^−1^ (amide II), 1284 cm^−1^ (amide 3), and 1278 cm^−1^ (C-O asymmetric vibration), which are consistent with the findings of other authors [41]. The PMMA results showed peaks at 1722 cm^−1^ (carbonyl groups), 1064 cm^−1^ (vibrations of the -C-O-C- bond), and around 1248 cm^−1^ (stretching vibrations of the C-O bond) [42]. When analyzing the spectrum for PDA-coated PMMA, peaks at 3342 cm^−1^ and 1245 cm^−1^ from the dopamine and peaks at 1722 cm^−1^ and in the range between 1064 cm^−1^ and 1440 cm^−1^ were observed, which confirms that polydopamine effectively coated the PMMA substrate.

However, conflicting results regarding a reduction in the adhesion of *S. mutans* to the coated or patterned surfaces, regardless of the PDA concentration (2 mg/mL or 10 mg/mL), were found. This finding may have different explanations and a variety of factors may have played a significant role.

Firstly, the hydrophobicity was drastically reduced after coating or patterning the PMMA surfaces with PDA. *C. aurea* showed a CA > 150°, which classifies it as a super hydrophobic surface according to Kim and Choi [43] and Falde et al. [44]. According to Zhang et al. [45], a high hydrophobicity reduces the surface energy and the microorganisms’ adhesion. The CA values for the PDA-patterned surfaces at both concentrations were considerably reduced to the point of considering them hydrophilic (~70° to 80°). The PDA-coated surfaces at both concentrations were even more hydrophilic (~26° to 34°). It has been demonstrated that PDA, as a super hydrophilic material, reduces the hydrophobicity of different surfaces, such as polyethylene, poly(vinyl fluoride), and polytetrafluoroethylene [46,47,48,49], among others, which was also observed for PMMA in the current investigation. Therefore, since *S. mutans* has shown a hydrophilic behavior and a predilection to adhere to more hydrophilic surfaces, as demonstrated by Satou et al. [50], the increased adhesion of *S. mutans* to the most hydrophilic surfaces (PDA-coated) observed in the current investigation may be explained. In addition, when comparing the adhesion to PDA-modified surfaces (coated or patterned) with the adhesion to PMMA alone (control group), similar values were found for PMMA and PDA-patterned surfaces, and such values were higher than those for PDA-coated surfaces. From a hydrophobicity perspective, the higher hydrophobicity of PMMA and PDA-patterned surfaces may account for the similarities in bacterial adhesion to both surfaces.

Secondly, even though there were no statistically significant differences in the adhesion of *S. mutans* to the control or patterned surfaces, patterning with PDA did show a small effect. When comparing PDA coating versus patterning at 2 mg/mL and 10 mg/mL, a lower adhesion was found to the patterned surfaces. However, the effect was not as strong, possibly due to the fact that natural surfaces, such as leaves, possess chemical coats in the form of waxes, as well as hierarchical structures, that cannot be transferred to artificial surfaces using soft lithography [51,52,53]. Therefore, the self-cleaning effect observed on natural surfaces could not be observed in the PDA-patterned surfaces. It is relevant to point out that no other antibacterial chemical compounds were attached to PDA in the current investigation, and it was used only to coat or pattern the surface, which falls under the category of physical surface modification. Furthermore, fluorescence microscopy showed more damaged and dead bacterial cells on the patterned surfaces. Hochbaum and Aizenberg [35] and Chung et al. [39] hypothesized that physical features on the surfaces of a material may act as obstacles for bacteria to adhere and organize properly, but this does not explain why patterning the surface destroyed more bacteria than coating it with the same compound (PDA). The explanation for this finding remains elusive. 

PDA, however, did exhibit a strong antibacterial effect when used directly on planktonic *S. mutans* cells. The antibacterial effect of PDA has been associated with different mechanisms, including the destruction of the bacterial cell membrane by chelating ions or proteins [46] or the generation of reactive oxygen species (ROS) that destroy the bacterial cell wall [54]. Protocols using PDA established an antibacterial and antifungal minimum inhibitory concentration (MIC) of 2 mg/mL [55]. In the current investigation, different PDA concentrations (2 mg/mL to 10 mg/mL) were tested to identify the concentration that showed the highest antibacterial effect on *S. mutans*. Even though all concentrations showed an antibacterial effect, the higher the concentration, the stronger the antibacterial effect. Concentrations of 8 mg/mL and higher showed a stronger antibacterial effect than chlorhexidine. Nonetheless, when the MIC and the highest inhibitory concentration (10 mg/mL) were used to coat or pattern the PMMA discs, this strong inhibitory effect was not as clear. As stated above, the hydrophobicity and topography-transferring restrictions might have played a more significant role. No differences in the roughness were observed among all the evaluated surfaces, and this surface characteristic did not play a substantial role in the current investigation, which contrasts the results by De la Pinta et al. [56]. In addition, Buergers et al. [6] could not confirm a correlation between bacterial adhesion and the roughness or hydrophobicity of dental PMMA. Further studies should test other bacterial strains that do not show a strong adhesion to hydrophilic surfaces, as well as other substrates, to observe whether the results of the current investigation can be replicated with different bacterial species/substrate scenarios. In addition, more investigations are needed to understand the damaging capabilities shown by patterned surfaces versus smooth-coating and whether different topographies would show similar results with other bacterial species or microorganisms in general.

## 5. Conclusions

Within the limitations of the current investigation, PDA exhibited an important antibacterial effect against *S. mutans* when used on planktonic cells from this bacterial species. This antibacterial activity could not be confirmed when the PMMA surfaces were coated or patterned with different concentrations of PDA, although patterning did show a better effect at reducing the adhesion of *S. mutans* than smooth-coating. This might be related to the material’s surface properties that were not addressed in the current investigation and that may play a more significant role in the adhesion mechanisms of this bacterial species to PMMA. It is essential to illustrate that PDA in the current investigation was not used to anchor other antibacterial chemical compounds, but to physically modify the surface of PMMA in the form of a smooth coat or a pattern. It is important, however, to emphasize that the promising results obtained with PDA in solution open the possibility of continuing the investigation of the effect of this compound on other materials and microorganisms. 

## Figures and Tables

**Figure 1 pathogens-12-01223-f001:**
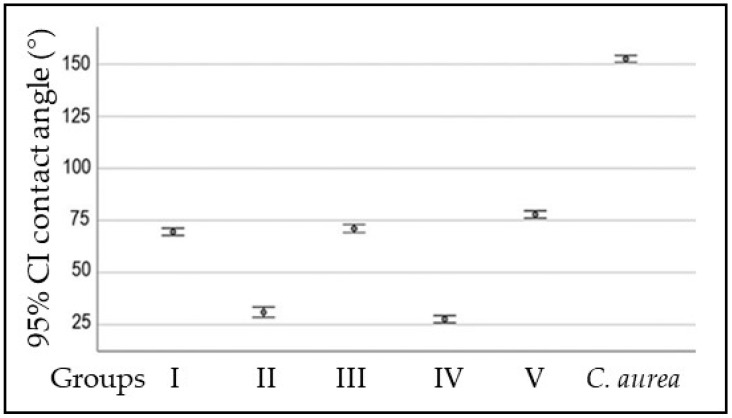
Contact angle (°) for experimental and control surfaces, as well as the *C. aurea* leaf. Data are shown as the average ± SD.

**Figure 2 pathogens-12-01223-f002:**
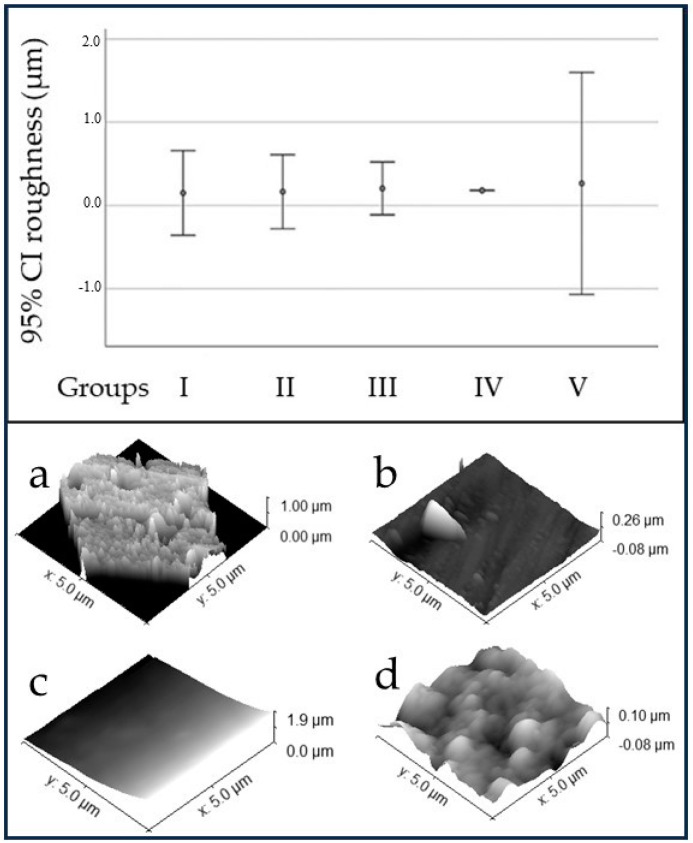
Roughness (µm) of PMMA surfaces (**top**). AFM images of *C. aurea* (**a**), polished PMMA (control group, (**b**)), smooth PDA-coated PMMA (group II, (**c**)), and PDA-patterned PMMA (group V, (**d**)).

**Figure 3 pathogens-12-01223-f003:**
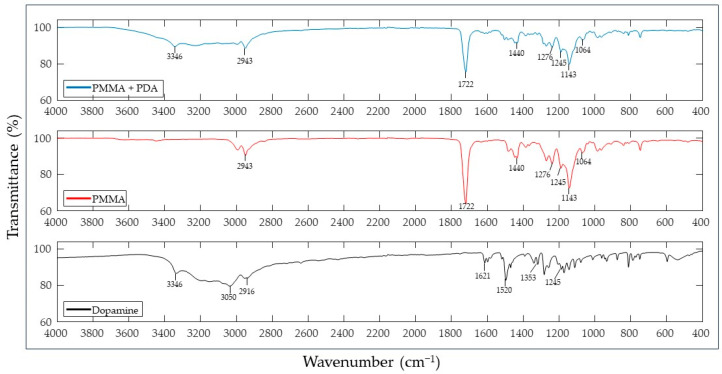
FTIR spectra of PDA-coated PMMA (**top**), PMMA (**middle**), and dopamine (**bottom**).

**Figure 4 pathogens-12-01223-f004:**
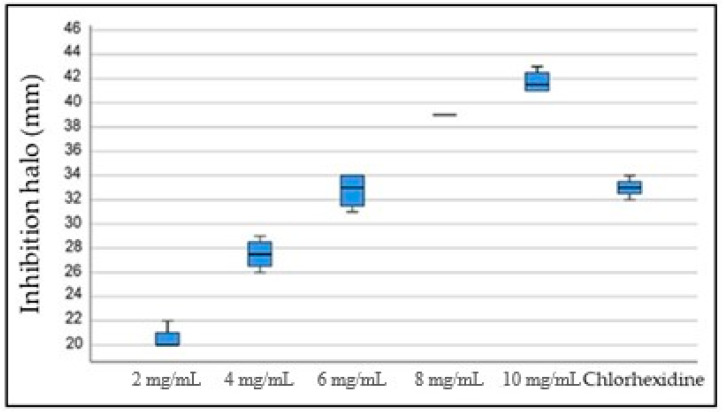
Inhibitory effect of the different PDA concentrations and 0.2% chlorhexidine on *S. mutans* ATCC 25175.

**Figure 5 pathogens-12-01223-f005:**
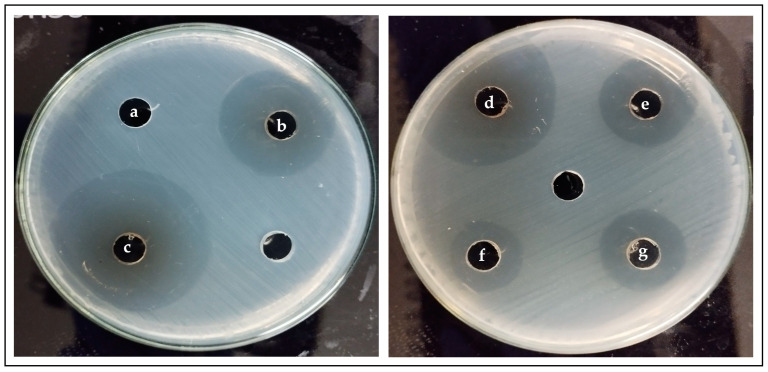
Inhibition haloes. (**a**) Negative control (buffer), (**b**) 6 mg/mL PDA, (**c**) 10 mg/mL PDA, (**d**) 8 mg/mL PDA, (**e**) 4 mg/mL PDA, (**f**) 2 mg/mL PDA, and (**g**) positive control (chlorhexidine). Unmarked spots in both images were not subjected to any treatment.

**Figure 6 pathogens-12-01223-f006:**
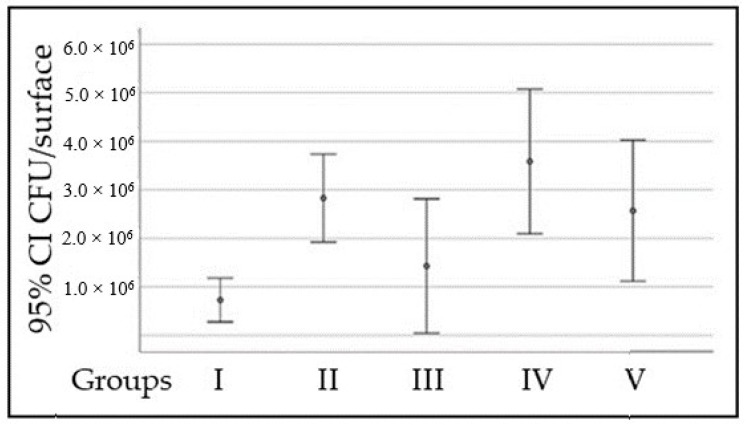
*S. mutans* adhesion to the surface of PMMA discs.

**Figure 7 pathogens-12-01223-f007:**
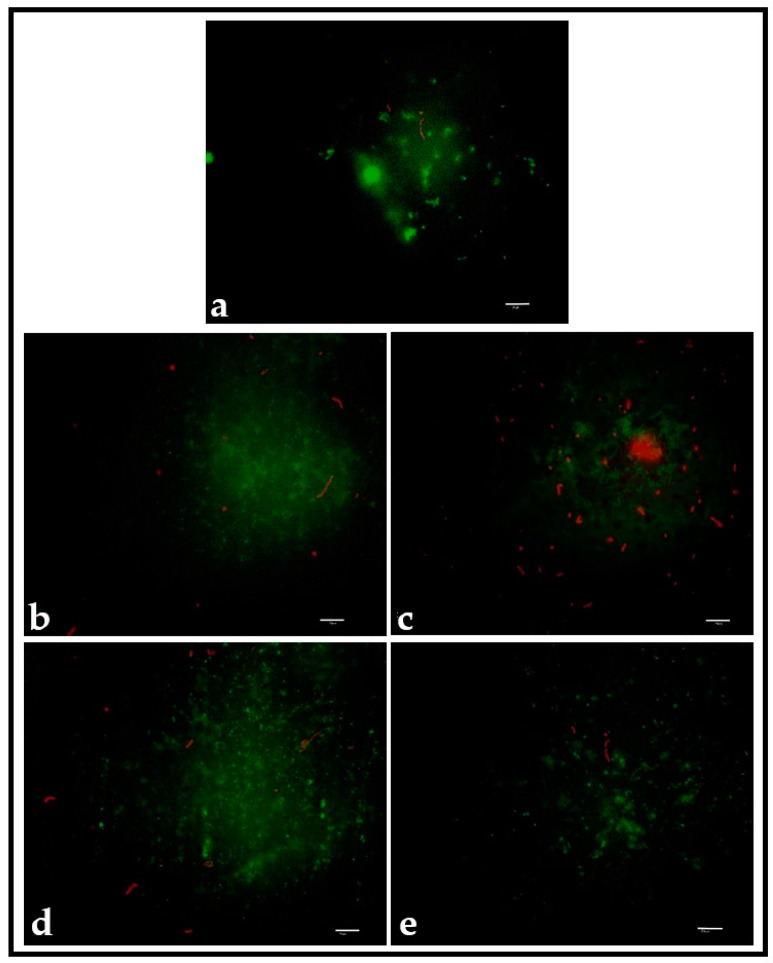
DEAD/LIVE staining of control group (**a**), coating and patterning with 2 mg/mL PDA ((**b**,**c**), respectively), and coating and patterning with 10 mg/mL PDA ((**d**,**e**), respectively). Scale bar is 10 µm.

**Table 1 pathogens-12-01223-t001:** Contact angle (°) for experimental and control surfaces, as well as the *C. aurea* leaf.

Surface	Contact Angle (°)	*p* Value
PMMA	(Median ± SD)	˂0.001
Group I	69.56 ± 2.46	
Group II	30.96 ± 3.44	
Group III	71.11 ± 2.71	
Group IV	27.61 ± 2.44	
Group V	77.88 ± 2.46	
*C. aurea*	152.59 ± 1.96	

One-factor ANOVA. Group I: polished PMMA discs (control); group II: PMMA discs dip-coated in 2 mg/mL PDA; group III: PMMA discs patterned with 2 mg/mL PDA; group IV: PMMA discs dip-coated in 10 mg/mL PDA; and group V: PMMA discs patterned with 10 mg/mL PDA.

## Data Availability

The data presented in this study are available from the corresponding author upon request.

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
