# Peer review of "Assessment of Polydopamine to Reduce Streptococcus mutans Adhesion to a Dental Polymer"

_pathogens, 2023, doi:10.3390/pathogens12101223_

Round 1

Reviewer 1 Report

The manuscript investigates the anti-adhesive efficacy of polydopamine-coated PMMA surfaces and PDA-PMMA surfaces combined with a biomimetic approach to reduce S. mutans adhesion on dental biomaterials. I think this work is meaningful, while some parts need to be clarified.

 -          Surfaces containing PDA did not significantly reduce the number of S. mutans adhered cells compared to PMMA control surfaces, although PDA shows promising antibacterial activity. Please relate these findings with the surface characterization data (e.g. surface hydrophobicity)

 -          According to LIVE/DEAD staining results (figure 6), biomimetic surfaces containing PDA seem to display much more damaged cells than PDA-coated PMMA surfaces. Did the authors analyze the reason?

-          Additional experiments should be adopted to demonstrate the presence of polydopamine coating on PMMA surfaces.

Reviewer 2 Report

1) The authors should add an AFM scan of the surface roughness of the mastermold (leaf in this case) and that of the PDMS stamp fabricated with it.

2) The spin coating parameters for PMMA on substrates should be mentioned.

3) Figure 6 does not reveal any strong difference between the coating and patterning of the surface. Either the figures should be improved or an AFM scan of coated and patterned surface should be added. 

4) The article clearly misses the effect of toxicity being induced by PMMA. A rationale for use of PMMA would be non-toxic in this case after PDA coating should be added.

Round 2

Reviewer 1 Report

The authors have addressed all suggestions made. I only have a doubt about line 318: Did the authors mean 'lower than PDA-coated surfaces'?

Reviewer 2 Report

The authors have very well answered the to the questions mentioned in the review process. 

Author Response

Thank you very much for your comments and suggestions, they improved the quality of this work.

Regards